# Metabolic Profiling and Comparative Proteomic Insight in Respect of Amidases during Iprodione Biodegradation

**DOI:** 10.3390/microorganisms11102367

**Published:** 2023-09-22

**Authors:** Pamela Donoso-Piñol, Gabriela Briceño, Joseph A. M. Evaristo, Fábio C. S. Nogueira, Barbara Leiva, Claudio Lamilla, Heidi Schalchli, María Cristina Diez

**Affiliations:** 1Doctoral Program in Science of Natural Resources, University of La Frontera, Temuco 4780000, Chile; p.donoso01@ufromail.cl (P.D.-P.); barbara.leiva@ufrontera.cl (B.L.); 2Department of Chemistry Science and Natural Resources, University of La Frontera, Temuco 4780000, Chile; 3Biotechnological Research Centre Applied to the Environment (CIBAMA-BIOREN), University of La Frontera, Temuco 4780000, Chile; claudio.lamilla@ufrontera.cl (C.L.); heidi.schalchli@ufrontera.cl (H.S.); 4Laboratory of Proteomics, LADETEC, Institute of Chemistry, Federal University of Rio de Janeiro, Rio de Janeiro 22775-000, Brazil; joseph.am.evaristo@gmail.com (J.A.M.E.); fabiocsn@ufrj.br (F.C.S.N.); 5Department of Chemical Engineering, University of La Frontera, Temuco 4780000, Chile

**Keywords:** pesticide degradation, amidase, proteome, pesticide-tolerant bacteria

## Abstract

The fungicide iprodione (IPR) (3-(3,5-dichlorophenyl) N-isopropyl-2,4-dioxoimidazolidine-1-carboxamide) is a highly toxic compound. Although IPR has been restricted, it is still being applied in many places around the world, constituting an environmental risk. The biodegradation of IPR is an attractive option for reducing its residues. In this study, we isolated thirteen IPR-tolerant bacteria from a biopurification system designed to treat pesticides. A study of biodegradation using different strains was comparatively evaluated, and the best degradation rate of IPR was presented by *Achromobacter* sp. C1 with a half-life (T_1/2_) of 9 days. Based on a nano-LC-MS/MS analysis for the strains, proteins solely expressed in the IPR treatment were identified by highlighting the strain *Achromobacter* sp. C1, with 445 proteins primarily involved in the biosynthesis of secondary metabolites and microbial metabolism in diverse environments. Differentially expressed protein amidases were involved in six metabolic pathways. Interestingly, formamidase was inhibited while other cyclases, i.e., amidase and mandelamide hydrolase, were overexpressed, thereby minimizing the effect of IPR on the metabolism of strain C1. The dynamic changes in the protein profiles of bacteria that degrade IPR have been poorly studied; therefore, our results offer new insight into the metabolism of IPR-degrading microorganisms, with special attention paid to amidases.

## 1. Introduction

Pesticides are chemicals used to control, prevent, or destroy pests that affect the health of crops [1]. Iprodione (IPR) is an efficient dicarboximide fungicide that was first manufactured in the 1970s, mainly to control fungal strains of the genera *Botrytis*, *Sclerotinia*, and *Monilinia* [2]. Currently, IPR and other dicarboximide pesticides are used principally in fruticulture and horticulture [3]. IPR is characterized as having moderate persistence in the soil, with a half-life (T_1/2_) from 7 to 60 days; it is slightly mobile on soil due to the low sorption coefficient (K*_oc_*) reported, which can fluctuate between 426 and 700 [4]. Despite IPR being recognized for its low leachability (GUS index of 0.43), loss of IPR can occur at concentrations exceeding water quality limits for aquatic ecosystems [5]. The primary degradation product of IPR is 3,5-dichloroaniline (3,5-DCA), which is recognized to be more toxic than its parent compound [6], while both compounds are considered potential carcinogens [2,7]. In addition, some studies have demonstrated that IPR has a strong negative impact on soil microorganisms. The high accumulation and persistence of IPR in soils could induce changes in the structure, diversity, and functionality of the soil and rhizosphere microbiome [8,9]. For these reasons, IPR has been banned in the European Union and use-limited in other countries but is still in use in Latin America [10,11].

Microbial biodegradation is the main mechanism responsible for the removal of IPR. Therefore, innovative tools need to be developed to biodegrade IPR and its metabolite to counteract its negative impact on the environment and human health. Soil bacteria, isolated from soils exhibiting accelerated pesticide degradation, have been used as a source of IPR-degrading microorganisms with a high capacity to degrade both IPR and 3,5-DCA [7,12]. Studies have confirmed that IPR can be biodegraded to metabolite I and isopropylamine by *Arthrobacter* sp. C1; then, it can be degraded to metabolite II by *Achromobacter* sp. C2; finally, *Arthrobacter* sp. C1 can produce the metabolite 3,5-DCA and glycine [13] (Figure 1). 

A biopurification system (BPS) for pesticide removal is based on adsorption and degradation processes occurring in an organic biomixture composed of soil, peat, and wheat straw. There are many diverse microorganisms in the organic biomixture of a BPS, with the microbial community composition mainly comprising the phyla Proteobacteria, Firmicutes, Bacteroidetes, and Actinobacteria [14,15]. Recent studies have reported that the bacteria with the greatest presence in a BPS are *Pseudomonas nitroreducens* [16], which may be the primary cause of the degradation of pesticides such as atrazine, carbofuran, and glyphosate. In addition, *Achromobacter* sp. strain C1 and *Pseudomonas* sp. strain C9, isolated from a BPS, were described as being responsible for chlorpyrifos, atrazine, and IPR removal from a liquid medium [15]. Although a BPS is recommended for the mitigation of different pesticides commonly applied on farms, this biopurification system also makes it possible to obtain bacteria with a high pesticide removal potential in different environmental matrices. Pesticide exposure increases bacterial diversity in the organic biomixture of a BPS [14], increases microbial activity [16], and favors the presence of diverse enzymatic responses, including acid and alkaline phosphatases, valine amido peptidases, lipases, and amylases, among others [15]. 

Amidases or amido hydrolases (3.5.1.X) comprise a large group of enzymes capable of hydrolyzing the C-N bonds of various amide compounds with the production of corresponding carboxylic acids [17]. Amidases are known to be important in the metabolism of xenobiotics, including pesticides [18,19,20]. These enzymes have been identified in various organisms, including bacteria, yeasts, fungi, plants, and animal tissues. Chacko et al. [21] reported a *Pseudomonas putida* MTCC 6809 rhizobacteria that produces amidase. Chen et al. [22] reported a novel amidohydrolase (DmhA) from *Sphingomonas* sp. that is able to hydrolyze the organophosphorus insecticide dimethoate to dimethoate carboxylic acid and methylamine. Yang et al. [23] showed that the amidase enzyme gene ipaH was responsible for the initial step of IPR degradation by *Paenarthrobacter* sp. strain *YJN-G* isolated from soil samples collected from IPR-applied vineyards. 

The potential application of amidases in biodegradation and bioremediation has received considerable interest in recent years [17]. However, proteomics approaches toward understanding the cellular and molecular mechanisms involved in IPR removal have rarely been investigated. This analysis helps to provide a comprehensive picture of the physiological responses and metabolism of IPR in bacteria isolated from BPS that can tolerate and remove IPR. Therefore, in this study, we sought to examine IPR removal by bacteria isolated from a BPS and identify bacterial IPR-degrading amidases using comparative proteomic approaches.

## 2. Materials and Methods

### 2.1. Chemicals and Media

Analytical standards of IPR and 3,5-DCA (purity 99%) were purchased from Sigma-Aldrich (St. Louis, MO, USA). For the removal assay, formulated commercial IPR (Rovral 50 WP) was purchased from Agan Chemical Manufacturers Ltd. A stock solution of 10,000 mg L^−1^ of IPR was dissolved in dimethyl sulfoxide, filtered through a 0.22 µm polytetrafluoroethylene (PTFE) filter, and then stored at 4 °C until use. All other chemicals and solvents were of analytical reagent grade (Merck-Sigma, St. Louis, MO, USA).

The media used for microbiological procedures were as follows: mineral salt medium (MSM) containing (per liter) 1.6 g K_2_HPO_4_, 0.4 g KH_2_PO_4_, 0.2 g MgSO_4_ ∗ 7H_2_O, 0.1 g NaCl, 0.02 g CaCl_2_, and 1 mL salt stock solution (2.0 g boric acid, 1.8 g MnSO_4_ ∗ H_2_O, 0.2 g ZnSO_4_, 0.1 g CuSO_4_, 0.25 g Na_2_MoO_4_, and 1000 mL distilled water); Luria Bertani (LB) broth containing (per liter) 5.0 g NaCl, 5.0 g yeast extract, 10.0 g tryptone, and 1000 mL distilled water; Luria Bertani modified broth (LB modified) containing (per liter) 2.5 g NaCl, 2.5 g yeast extract, 5.0 g tryptone, and 1000 mL distilled water; International Streptomyces Project-2 Medium (ISP2) containing (per liter) 10.0 g malt extract, 4.0 g malt extract, 4.0 g glucose, and 1000 mL distilled water; and Reasoner’s 2A agar (R2A) containing (per liter) 0.5 g casein acid hydrolysate, 0.5 g yeast extract, 0.5 proteose peptone, 0.5 g glucose, 0.5 g soluble starch, 0.3 g K_2_HPO_4_, 0.024 g MgSO_4_, and 0.3 g sodium pyruvate. When required, the culture media were solidified by adding 15 g of agar. The media were autoclaved at 121 °C, and the pH was adjusted to 6.5. 

### 2.2. Sample Collection and Isolation of IPR-Degrading Bacteria

The bacterial strains were isolated from the organic biomixture sample obtained from a BPS used in recent years for the treatment of atrazine, chlorpyrifos, and IPR. The BPS consisted of a plastic tank of 1 m^3^ capacity packed with 125 kg of a biomixture prepared with topsoil, commercial peat, and wheat straw at a ratio of 1:1:2 (*v v*^−1^), which reached a height of 60 cm with a water-holding capacity of 60–70% [15]. Biomixture samples were taken across the BPS and transported to the laboratory in sterile plastic bags. The samples were homogenized and stored at 4 °C overnight. 

The isolation of bacterial strains was carried out using direct and enrichment methods as described by Campos et al. [12] with modifications. For direct isolation, 10 g of the biomixture sample was suspended and homogenized in 90 mL of sterile physiological solution (0.9% NaCl), serially diluted (10^−1^–10^−5^), and an aliquot of 10 μL was cultivated on R2A, ISP2, and LB agar medium. For enrichment isolation, 10 g of a biomixture was suspended in flasks containing 90 mL of MSM supplemented with 10 mg L^−1^ of IPR; incubation was performed at 28 °C and 130 rpm under constant shaking for seven days. After incubation, an aliquot was taken and transferred sequentially after a new incubation period into new flasks containing 25, 50, and 100 mg L^−1^ of IPR. The last incubation period was finalized, serial dilution (10^−1^–10^−5^) was performed, and aliquots of 10 μL were plated on R2A, ISP2, and LB agar medium. All plates were cultivated at 28 °C for seven days. Bacterial colonies with different morphological characteristics were purified and maintained as pure cultures in LB agar medium slant at 4 °C. 

### 2.3. Inoculum Preparation 

Starter cultures of pure bacterial isolates were grown in a 100 mL flask containing 50 mL of LB diluted broth adjusted to pH 6.5. This medium was selected because the bacterial strains grow better compared with MSM supplemented with IPR as the sole carbon source [15]. Two strains identified as *Achromobacter* sp. strain C1 (MK110041) and *Pseudomonas* sp. strain C9 (MK110046), which were previously isolated from a BPS and characterized to be able to degrade chlorpyrifos and IPR [15], were also included in this study. The inoculated flasks were incubated for 24 h at 28 °C and 130 rpm in a rotary shaker. The cultures were centrifuged (6500 rpm for 10 min at 4 °C), and the cell pellets were resuspended in a sterile physiological solution to obtain a cell density (OD_600_) of 1.0. 

### 2.4. Screening of IPR-Degrading Bacteria 

An inoculum of each bacterial strain was incubated in a 200 mL flask containing 50 mL of diluted LB broth supplemented with 50 mg L^−1^ IPR. For the experiments, biomass obtained during inoculum preparation (see Section 2.3) was used at a concentration of 1% (*v v*^−1^). The cultures were incubated in an orbital shaker at 130 rpm and 28 °C for 96 h under dark conditions. Then, an aliquot (1 mL) was collected and centrifuged (6500 rpm for 10 min at 4 °C), and 0.5 mL of supernatant was diluted in 1 mL of chromatographic-grade acetonitrile. After sample preparation and filtration through a 0.22 µm PTFE membrane, the residual IPR concentrations determined via HPLC were measured as mentioned below. Non-inoculated flasks and inoculated flasks without IPR were used as abiotic and biotic controls, respectively. 

### 2.5. Characterization and Identification of IPR-Degrading Bacterial Strains

Biochemical characterization using the APIZYM kit (Biomerieux, Marcy l’Etoile, France), extracellular hydrolytic enzyme production, and cellular visualization via scanning electron microscopy (SEM) were carried out as described in Briceño et al. [15].

The genetic material of the selected bacterial isolates was extracted using the DNeasy UltraClean© Microbial Kit (Qiagen, Hilden, Germany) according to the manufacturer’s instructions. The housekeeping gene 16S rRNA of the bacterial isolates was amplified via polymerase chain reaction (PCR) mixtures that contained 100 ng of bacterial genetic material, GoTaq^®^ Green Master Mix 1X (Promega, Madison, WI, USA), and 0.2 µM of universal forward primer 27 F (3′–AGAGTTTGATCMTGGCTCAG–5′) and universal reverse primer 1492R (3′–TACGGYTACCTTGTTACGACTT–5′); a final volume of 50 µL was adjusted with nuclease-free water [24,25]. The PCR program included an initial denaturing step at 94 °C for 5 min, followed by 30 cycles of denaturing for 30 s at 94 °C, annealing for 1.5 min at 50 °C, and extension at 72 °C for 1.5 min. A final extension step of 72 °C for 10 min was performed. Amplimers of 16S rRNA (1.5 kb) were visualized in 1% agarose gel containing GelRed^®^ staining 3X (Merk Millipore, Temecula, CA, USA). Then, 16S rRNA fragments were sequenced by Macrogen Corporation (Seoul, Republic of Korea). The obtained DNA sequences were compared with existing sequences in GenBank using the BLAST tool (BLAST^®^ NCBI, Bethesda, MD, USA). The phylogenetic affiliation in relation to representative dicarboximide-degrading bacteria in GenBank was performed using MEGA v7.0 software [26]. To evaluate the reliability of the tree, 1000 bootstrap replications were performed. 

### 2.6. Bacterial Growth and IPR Removal

Selected and identified strains were inoculated (1% *v v*^−1^) in conical flasks with 300 mL of diluted LB media supplemented with 50 mg L^−1^ of IPR. Flasks were cultivated under dark conditions on a rotary shaker at 130 rpm for 48 h at 28 °C. Liquid samples were obtained at 3, 6, 9, 12, 18, 24, 36, and 48 h of incubation. Biomass growth was quantified by absorbance measurement at 600 nm and converted to cell dry weight (g L^−1^) through a calibration curve [27]. Then, the supernatant (0.5 mL) diluted in acetonitrile was filtered, and the concentrations of IPR and the main metabolite 3,5-DCA were determined via HPLC according to the method described by Diez et al. [28]. The limit of quantification (LOQ) and limit of detection (LOD) were 0.238 mg L^−1^ and 0.089 mg L^−1^ for IPR, respectively. For 3,5-DCA, the LOQ was 0.236 mg L^−1^, and the LOD was 0.076 mg L^−1^. 

### 2.7. Proteome Preparation, Digestion, and Mass Spectrometry Analysis

IPR-degrading strains selected in a procedure similar to that described in Section 2.6 were exposed to 50 mg L^−1^ of IPR for 24 h. At this time, cell biomass was harvested by centrifugation at 8000× *g* for 20 min at 4 °C and then washed twice with phosphate-buffered saline (PBS) containing (per L) 8.0 g NaCl, 1.15 g Na_2_HPO_4_, 0.2 g KCl, and 0.2 g KH_2_PO_4_, adjusted to pH 7.0. After centrifugation, the pellet was resuspended in 5 mL PBS buffer containing 25 µL of 1 mM of phenylmethylsulphonyl fluoride (PMSF) used as a protease inhibitor. Then, the cells were disrupted via ultrasonication on ice and centrifuged at 8000× *g* for 15 min at 4 °C. The supernatant was then obtained and lyophilized for its conservation. 

For desalting, the lyophilized sample was resuspended in 200 μL of water MilliQ© (Merck, Darmstadt, Germany) and 800 μL of cold acetone and kept overnight at −30 °C. The sample was centrifuged in two cycles at 10,000× *g* for 30 min and 4 °C, the supernatant was discarded, and the extracts were dried at room temperature. The protein extracts were resuspended with 500 μL of ammonium bicarbonate (25 mM) and 250 μL of water MilliQ© and mixed. After sample preparation, the protein concentration was verified using the Qubit^TM^ 2.0 Fluorometer (Invitrogen, Carlsbad, CA, USA) according to the manufacturer’s instructions. After determining the protein concentration, 100 µg of the protein sample was established. An aliquot of 100 mM dithiothreitol was added to the sample to obtain a 10 mM final concentration. Next, the samples were homogenized and incubated at 800 rpm and 60 °C for 1 h. After incubation, an aliquot of 400 mM iodoacetamide was added to obtain a final concentration of 40 mM. Samples were incubated for 30 min in dark conditions. 

The trypsin digestion was carried out at a ratio of 1:100 *w w*^−1^ of protein and 50 mM ammonium bicarbonate buffer. For protein digestion, 75 μL of trypsin (equivalent to 0.02 µg) was added to the sample and incubated at 800 rpm and 37 °C for 20 h. Trypsinization was quenched by applying trifluoroacetic acid (TFA, 1%) to reduce the pH to approximately 2. Next, the sample was applied to stage tip columns made with Applied Biosystems™ POROS™ R2 resin (Thermo Fisher Scientific, Waltham, MA, USA) for cleaning up; the elution was made using a gradient of ACN (50 and 70%). Finally, the peptides were dried at 45 °C for 1 h using a SpeedVac concentrator (Thermo Fisher Scientific, Waltham, MA, USA), resuspended in 100 µL 0.1% formic acid, standardized, and quantified using the Qubit^TM^ 2.0 Fluorometer protein assay kit [29].

The samples were analyzed in technical triplicates using nano-liquid chromatography (Easy-nLC 1000, Thermo Scientific) coupled to a hybrid Quadrupole Orbitrap mass spectrometer (Q Exactive Plus, Thermo Scientific). The peptides from the samples were loaded in a home-made C18 trap column (2 cm × 150 μm internal diameter, 5 μm ReproSil-Pur resin, Dr. Maisch GmbH, Ammerbuch, Germany), and the peptides were separated using a home-made C18 New Objective PicoFRIT column (25 cm × 75 μm internal diameter, ReproSil-Pur resin, 3 μm Dr. Maisch GmbH; Ammerbuch, Germany). The chromatographic flow was 0.3 mL min^−1^, and a linear gradient was applied starting with 100% of mobile phase A (0, 1% formic acid, 5% ACN) and increasing to 40% of mobile phase B (0.1% formic acid, 95% ACN) in 180 min. The peptides were ionized and transferred using a nanoelectrospray source (Thermo Scientific, Waltham, MA, USA) with positive polarity, a potential of 3.0 kV, and heating at 250 °C.

The mass spectrometer was operated in data-dependent analysis mode with dynamic exclusion of 45 ms and full-scan MS1 spectra with a resolution of 70,000 at *m/z* 200, followed by fragmentation of the 15 most intense ions with high collisional dissociation (HCD), normalized collision energy (NCE) of 30, and resolution of 17,500 at *m/z* 200 in MS/MS scans. Species with a charge of +1 or greater than +4 were excluded from MS/MS analysis. 

### 2.8. Data Analysis

#### 2.8.1. Bacterial Growth and IPR Degradation Analysis

The data obtained in the assay described in Section 2.6 were used to determine the specific growth rate (µ) by plotting ln(B_t_/B_0_) against time, where B_0_ is the amount of biomass in liquid medium at time zero and B_t_ is the amount of biomass at time t. The biomass duplication time was determined as ln(2)/µ. The IPR removal percentage was calculated using the following equation: (C_i_ − C_f_/C_i_) × 100, where C_i_ is the initial concentration and C_f_ is the final concentration. The removal kinetic constants using the first-order kinetic model were calculated using the following equation: ln(C_t_/C_0_) = e^−kt^, where k is the degradation rate constant (h^−1^), C_t_ is the IPR concentration (mg L^−1^) at time t (h), and C_0_ is the IPR concentration (mg L^−1^) at time 0 (h). The time at which IPR concentrations were reduced by 50% (T_1/2_) was calculated using the following equation: T_1/2_ = ln(2)/k.

#### 2.8.2. Proteomic Data Analysis

Mass spectrometric data were analyzed using Proteome Discoverer 2.1 software (Thermo Scientific). Identification of peptides via peptide spectrum matching was performed with the Sequest HT algorithm against a protein database comprising the common contaminants in mass-spectrometry-based proteomics analysis concatenated with the *Achromobacter spanius* (A0A2S0IDW1; Taxon ID 217203) available from UniProt http://www.uniprot.org/ (accessed on 10 December 2022). The searches were run with a peptide mass tolerance of 10 ppm, MS/MS of 0.05 Da, cleavage by trypsin, a maximum limit of two missed cleavages, fixed modification of carbamidomethyl, and variable modification of acetylation of the protein N-terminus and methionine oxidation. False discovery rates were obtained using the Percolator node selecting identification with a *q*-value equal to or less than 0.01 at peptide and protein levels. Protein quantification was performed using an extracted ion chromatogram (XIC) via the Precursor Ions Area Detector tool. The top 3 method (average of the 3 most intense peptides) was used for protein quantification, selecting peptides that were considered unique.

#### 2.8.3. Pathway Analysis

The identified proteins were then analyzed through BLAST2Go software version 4.1 [30] with the NCBI data bank [31] to map the proteins in diverse possible pathways. The KEGG© database was used to visualize the proteins on the metabolic pathway of the selected strain under IPR exposure using KEGG© Mapper visualization of the overexpressed and underexpressed proteins and proteins present solely in the presence of IPR.

### 2.9. Statistical Analysis

All experiments were performed in triplicate, and the standard deviation was calculated for the protein expression. The ratio between quantitative values from IPR exposure (B) and control (A) was determined in triplicate for each protein, and the mean was used to calculate the fold change (FC, B/A). To normalize the data, the fold change value was calculated with log_2_. To determine the significant values, the log_2_FC—median was calculated, and values within the [−SD–+SD] range of the median standard deviation were considered significant.

## 3. Results

### 3.1. Isolation and Selection of IPR-Degrading Bacteria

In this study, 83 different colonies were isolated from a BPS. Of these, 13 bacterial strains were able to grow and remove IPR. The results show that IPR removal by the new isolated strains and the two additionally included strains *Achromobacter* sp. C1 and *Pseudomonas* sp. C9 ranged between 32.0% and 97.3%, with 3,5-DCA metabolite production ranging from 0.043 mg L^−1^ to 0.930 mg L^−1^ after 96 h of incubation (Appendix A). Of the 15 strains evaluated, we selected for further study those that were able to remove over 96% of the IPR. These strains were strain I8, isolated using the direct isolation method, and strains R9, R14, and R57. In addition, we selected strains C1 and C9, which were obtained using an IPR-enrichment isolation method. 

### 3.2. Characterization and Molecular Identification

This procedure was conducted for the four new isolates tolerant to IPR. These were characterized based on some phenotypic, biochemical, and molecular characteristics. The morphological characteristics of bacteria evaluated via SEM showed that all strains have a bacillus cell shape with a size of 0.772 to 3.510 µm (Appendix A). Strains R9, R14, and R57 were Gram-negative, while strain I8 was Gram-positive. Except for strain R14, the strains were positive for amylase, cellulase, lipase, protease, and/or gelatinase activity (Appendix A). A further characterization conducted using api^®^ ZYM showed that all the strains were positive for esterase-(C4), esterase lipase (C8), leucine arylamidase, acid phosphatase, and naphthol-AS-BI-phosphohydrolase. Enzymes such as alkaline phosphatase and valine arylamidase were positive for all strains except for strains R14 and I8, respectively (Appendix A). 

The identification of selected strains via the 16S rRNA sequencing analysis showed that the bacteria belong to the phylum Firmicutes, family Bacillaceae, genus *Priestia* (strain I8), and phylum Proteobacteria, family Xanthomonadaceae, genus *Stenotrophomonas* (strain R9), and family Pseudomonadaceae, genus *Pseudomonas* (strain R14 and strain R57). A comparison of the 16S rRNA sequences (entire sequence compared with available sequences in GenBank) of strains I8, R9, R14, and R57 showed >99% similarity to *Priestia aryabhattai*, *Stenotrophomonas rhizophila*, *Pseudomonas vancouverensis*, and *Pseudomonas vancouverensis*, respectively (Appendix A).

The phylogenetic analysis based on the 16S rRNA was carried out for the four new isolates (Appendix A). The results indicate that strains R14 and R57 clustered with *Pseudomonas kilonensis* strain MB490 (MG685888.1) and *Pseudomonas* sp. strain C9. Strain I8 appeared to be phylogenetically clustered with *Bacillus thuringiensis* strain MB497 (KP886829.1). Finally, strain R9 was clustered with *Acinetobacter baumannii* strain DT (MN658561) and *Raoultella* sp. strain X1 (EU585749), and, together, they were associated with *Pseudomonas* sp. 

### 3.3. IPR Removal and Growth of IPR-Tolerant Bacteria

In this study, the four new isolates were evaluated for IPR removal, and strains *Achromobacter* sp. C1 and *Pseudomonas* sp. C9 isolated previously from a BPS were characterized as IPR-degrading strains [15]. Figure 2 shows IPR removal and 3,5-DCA production after 48 h in a liquid medium supplemented with 50 mg L^−1^ IPR and inoculated with the single selected strains. The IPR removal (Figure 2A) shows that at 12 h, strains C1, R9, and R14 showed the highest IPR removal, with values between 41 and 53% (related to the control treatment). At 24 h, IPR removal was between 58% and 91%, with strains C1, R14, and R57 exhibiting the highest IPR removal. Finally, at the end of the assay, the IPR removal for all strains achieved a removal rate of >87%, and strain C1 presented the highest IPR removal (95%), resulting in a final concentration of 1.5 mg L^−1^ IPR.

The kinetic data (Table 1) show that IPR removal by the strains was described by a rate constant of 0.047 h^−1^ to 0.076 h^−1^ resulting in a T_1/2_ value of 14.6 h to 9.1 h when strains I8 and C1 were inoculated in the medium, respectively. 

In addition to IPR removal, the production of 3, 5-DCA, its principal metabolite, was evaluated. The results show (Figure 2B) that 3, 5-DCA production was significantly different (*p* < 0.05) among the strains. The general trend observed was unaltered 3,5-DCA production for the control treatment. However, in the first 12 h, a mild increase in 3,5-DCA production was observed for all strains, with a concentration close to 0.25 mg L^−1^. After that, a metabolite increase was observed, with a final concentration of 0.40 mg L^−1^ when the liquid medium was inoculated with the strain *Achromobacter* sp. C1. 

The results of microbial growth (Figure 3) show that under IPR treatment, the biomass increased significantly after 6 h for all the strains except for strain I8, for which the biomass increased after 18 h under IPR treatment (Figure 3C). There was a slight difference between the microbial growth of strains R9, R14, and R57 under IPR and control treatment (Figure 3D–F) and a significant difference (*p* > 0.05) between both conditions for strain C9 (Figure 3B), where at 36 h, the biomass under IPR treatment presented a maximum value of 0.33 g L^−1^ in contrast to 0.16 g L^−1^ in the control. Finally, compared to the control treatment, the biomass of strain C1 was unaltered in the presence of IPR (Figure 3A). According to these results, microbial growth was described with a specific growth rate of 0.034 to 0.299 and a biomass duplication time of 2.3 to 39.1 h for the IPR treatment (Table 1). 

### 3.4. Comparative Proteomic Study

#### 3.4.1. Overall Findings

The present study focused on whole cell extracts to identify proteins associated with IPR degradation. It was carried out through a differential proteome compared with a control, profiling for all six selected strains exposed to the previous study at 50 mg L^−1^ IPR. For this, samples were harvested at 12 h when between 20 and 50% IPR removal was reached (Figure 2) and a significant increase in biomass was already observed (Figure 3). In this study, we found 1057, 1209, 1212, 1569, 1590, and 1740 total proteins in *Pseudomonas* sp. R57, *Stenotrophomonas* sp. R9, *Pseudomonas* sp. R14, *Priestia* sp. I8, *Pseudomonas* sp. C9, and *Achromobacter* sp. C1, respectively. The differential proteome profiling conducted for all of the strains showed several proteins with elevated (upregulated) and lowered (downregulated) levels of expression in response to IPR treatment (≥1.4 and ≥−1.4-fold), as shown in Figure 4. 

The values reported were from 118 to 189 upregulated proteins and a range of 51 to 153 downregulated proteins, except for strain I8, which showed 644 proteins with lower expression levels, representing about 41% of the strain I8-identified proteome. For the six strains, several proteins solely expressed (fold change infinite) in the IPR treatment were found. For five of the strains, the only expressed proteins ranged from 25 to 111 proteins, whereas strain C1 stood out with 445 proteins, representing about 36% of the strain C1-identified proteome. Considering these results in combination with the IPR degradation assay, we decided to delve more deeply into the *Achromobacter* sp. C1 strain proteome analysis.

#### 3.4.2. Analysis of Differentially Expressed Proteins of IPR-Treated and Untreated *Achromobacter* sp. C1

The proteome of *Achromobacter* sp. C1 exposed to 50 mg L^−1^ of IPR showed quantitative differences in expression compared to the untreated control. A total of 664 identified proteins for *Achromobacter* sp. C1 showed no differences between the IPR and control treatments. By contrast, 1076 proteins were differentially expressed, with 168 proteins being significantly upregulated (≥1.4-fold), 153 proteins significantly downregulated (≥−1.4-fold), and 445 proteins being solely expressed in the IPR treatment (fold change infinite) (Figure 4), indicating the induction of specific proteins/pathways that aid IPR degradation. The underlying molecular mechanism of IPR degradation was analyzed through the differentially expressed protein profiles. We found that the biological process involved in IPR degradation corresponded to the metabolic process and cellular process, representing 44% and 37%, respectively (Figure 5A). Additionally, based on the molecular function, the proteins identified in strain C1 were involved mainly in transferase activity (19%), oxidoreductase activity (17%), and hydrolase activity (16%) (Figure 5B).

The upregulated proteins analyzed using the KEGG database were involved in 17 different metabolic pathways. Those upregulated pathways are involved in the biosynthesis of secondary metabolites, ribosomes, microbial metabolism in a diverse environment, and biosynthesis of amino acids. Additionally, there were 11 downregulated proteins, the pathways of which are involved in microbial metabolism in various environments, the biosynthesis of secondary metabolites, and the biosynthesis of cofactors. Finally, the solely expressed proteins in the IPR treatment were involved in 37 metabolic pathways, with the biosynthesis of secondary metabolites, microbial metabolism in diverse environments, biosynthesis of amino acids, carbon metabolism, ABC transporter, propanoate metabolism, pyruvate metabolism, and valine, leucine, and isoleucine degradation being the key solely expressed pathways (Appendix A). 

### 3.5. Amidase Protein Expression of Achromobacter sp. C1 and Effect on Amidase Metabolic Pathways in Response to IPR

The differential proteome profiling in strains C1 identified amidase proteins with elevated/lowered expression levels in response to IPR. A list of proteins significantly upregulated under IPR exposure is provided in Table 2. The major upregulated proteins (≥1.4-fold) were aspartyl/glutamyl-tRNA (Asn/Gln) amidotransferase subunit C (A0A3Q9KM03), amidase (Uniprot ID A0A3Q9KKP8), and cyclase (A0A2S0I6B2). In addition, the major downregulated proteins were formamidase (A0A2S0I9W4) and amidase (A0A2S5GVY3). Finally, several proteins were solely expressed (fold-change infinite) during the treatment with IPR. Among these proteins, we found glutamyl-tRNA (Gln) amidotransferase subunit A (A0A3S4NGC5, A0A3S4PMM6, and A0A2S5GXA3), amidase (A0A3Q9KJB0 and A0A2S5GNM1), and an uncharacterized hydrolase YxeP with amidohydrolase activities (A0A448CBV6 and A0A3S4Q6N4), among other representatives of hydrolase activity. 

The KEGG pathway database analysis identified that amidase enzymes were involved in six metabolic pathways: phenylalanine metabolism, tryptophan metabolism, styrene degradation, aminobenzoate degradation, caprolactam degradation, and aminoacyl-tRNA biosynthesis (Figure 6). Exposure of strain C1 to IPR enhanced the expression of the protein in tryptophan metabolism, such as cyclase (CYC) involved in the pathway of nicotinamide metabolism and benzoate degradation through anthranilate formation. Furthermore, amidase (AMD) proteins were overexpressed in aminoacyl-tRNA biosynthesis with the synthesis of L-glutaminyl-tRNA and L-asparaginyl-tRNA. Moreover, the key stress-responsive proteins consist of formamidase (FOR), which is involved in 5 of 6 metabolic pathways. Specifically, FOR inhibition was observed for the phenylacetate and acrylate metabolite formation related to phenylalanine metabolism and styrene degradation metabolisms, respectively. Additionally, FOR protein was inhibited in aminobenzoate degradation and tryptophan metabolism pathways. However, this effect was accompanied by overexpression of the CYC protein and expression of new proteins solely expressed in IPR exposure, which were identified as uncharacterized hydrolase YxeP (U) and mandelamide hydrolase (MAN).

All of these enzymes are involved in anthranilate production, which is a precursor in nicotinamide metabolism and benzoate degradation. Furthermore, other enzymes such as AMD and MAN, expressed solely in the presence of IPR, were involved in 6-aminohexanoate dimer and cyclohexylamine, respectively, in caprolactam degradation metabolism. One AMD protein, together with glutamyl-tRNA Gln amidotransferase subunit A (GLN), was solely expressed in IPR exposure, and one upregulated AMD protein was involved in aminoacyl-tRNA biosynthesis. Finally, as shown in Table 2, in this study, we identified the proteins N-ethylammeline chlorohydrolase, N-formylglutamate deformylase, and N-formylglutamate amidohydrolase, with hydrolase and amidase activity described for the genus *Achromobacter* but without an identified metabolism pathway. 

## 4. Discussion

Diverse soil bacteria isolated from contaminated soils or soils with a history of pesticide application have been studied for IPR degradation [7,12,32,33]. The representative genera that can degrade this insecticide even at high concentrations (>100 mg L^−1^) include *Arthrobacter*, *Pseudomonas*, *Paenarthrobacter*, and *Microbacterium*, among others. In this research, 13 new bacterial strains were isolated from a routinely in-use BPS used for treating IPR, chlorpyrifos, and atrazine because these systems constitute a source of microbial strains adapted to tolerate and remove pesticides [15]. The characterization of the isolated bacteria made it possible to determine the production of different enzymes, many of them related to pesticide degradation. The production of a phosphomonoesterase enzyme like alkaline phosphatase was related by Briceño et al. [34] to the degradation of chlorpyrifos, probably using O-P bond hydrolysis. In addition, arylamidase enzymes have been reported as being involved in amide pesticide degradation via cleavage of the N-C bond [20,35]. Therefore, the biochemical characteristics of selected bacteria could be related to the ability to remove a high percentage of IPR (>96%), as observed in this study after 96 h of incubation. According to our results, the BPS system offers the acquisition and selection of potential isolates with a high tolerance to pesticides and degrading activity. 

Identification conducted via 16S rRNA gene sequencing showed a closer relation with the bacteria from the genera *Priestia* (previously *Bacillus*) [36], *Stenotrophomonas,* and *Pseudomonas,* which, in addition to *Achromobacter* (included in this study), are known as microorganisms that are capable of degrading different classes of pesticides, including methyl parathion, chlorpyrifos, dimethoate, atrazine, and IPR, among others [15,37,38]. This agrees with the phylogenetic analysis of isolates, which showed a closer relation with amide-moiety pesticide-degrading bacteria (Appendix A). 

In addition to the new BPS isolates, two previous strains identified by Briceño et al. [15] as *Achromobacter* sp. C1 and *Pseudomonas* sp. C9, which showed promising results for IPR degradation, were included as comparative strains. The six strains efficiently degraded IPR by over 87% in 48 h in LB diluted medium used instead of MSM medium due to the low microbial growth observed by us and other researchers [15,39]. The presence of a carbon source in addition to IPR could indicate possible IPR degradation by co-metabolism as hypothesized by Mercadier et al. [40] for IPR and by Zhao et al. [39] for β-cypermethrin. In addition, Campos et al. [13] reported that IPR transformation to metabolite I and metabolite II was only achieved in the presence of extra carbon and nitrogen sources. The IPR removal observed in this study was in agreement with previous reports: *Microbacterium* strains CQH-1 removed over 88% of 100 mg L^−1^ of IPR at 64 h [7], and the strains *Providencia stuartii* JD and *Brevundimonas naejangsanensis* J3 single and as a co-culture removed over 95% of different dicarboximide pesticides after six days, including 50 mg L^−1^ IPR [41]. In addition, the presence of metabolite 3,5-DCA confirmed the IPR removal, probably occurring through an enzymatic process where other metabolites not identified in this study might also be present. In relation to the initial mass of IPR, the amount in mass of 3,5-DCA represented on average 1.40% of the parent compound at the end of the study. Thus, the remaining percentage could be represented by the IPR intermediates identified as metabolite I, metabolite II, or even glycine, which is produced together with 3,5-DCA in the final stage of IPR degradation. However, more analyses are required to clarify this. The metabolite 3,5-DCA is recognized as the major metabolite of IPR degradation and dicarboximide fungicide in general. This compound is characterized to be more toxic and persistent than the parent compounds; therefore, biodegradation strategies must be developed to prevent its occurrence in high doses. In our study, *Achromobacter* sp. C1 was the most efficient degrader of IPR, but it also produced the highest amount of the toxic intermediate 3,5-DCA at the end of the assay. Therefore, to complement the efficiency of strain C1, using a co-culture able to contribute to the decrease in 3,5-DCA concentration over time could be investigated in a future study. Zhang et al. [41] reported that the co-culture formed by the strains *Providencia stuartii* JD and *Brevundimonas naejangsanensis* J3 could effectively degrade 3,5- DCA to phenol and then to muconic acid. 

The rapid removal of IPR by the six strains was evidenced by a decrease in the half-life times from 91.5 h in the control treatment to 9.1–14.6 h when bacteria were added to the medium. According to the USEPA [42], the aerobic aquatic degradation half-life of IPR is 9 days. However, treating liquids with IPR-degrading microorganisms can modify those times. Levio-Raiman et al. [43] observed IPR removal characterized by a T_1/2_ of 3.7 to 5.8 d after inoculating a single and microbial consortium of *Verticilium* sp. H5 and *Metacordyceps* sp. H12, with both fungi isolated from a BPS. Briceño et al. [15] reported an IPR T_1/2_ of 7 to 12 h for 50 mg L^−1^ of IPR in a liquid medium inoculated with different bacterial strains, including *Achromobacter* sp. C1 and *Pseudomonas* sp. C9. Different factors, including the class of microorganisms and culture conditions, may influence the difference in the removal rate and consequent half-life time for IPR. However, biomass growth is usually inhibited by the presence of pesticides, as reported for high concentrations (>400 mg L^−1^) [44,45]. This response can be countered by the addition of other carbon sources into the liquid culture medium rather than decreasing the concentration of the contaminant [46]. According to our observation, at the same IPR concentration, the presence of a double amount of nutrients can accelerate IPR removal, as we observed by comparing our results with those obtained by Briceño et al. [15], where a diluted LB medium was used. 

The overall comparative proteomic study conducted for the six strains to identify the response of proteins specifically associated with IPR metabolism resulted in 1057 to 1740 total proteins, representing between 21 and 31% of the predicted proteome for the bacterial strains. In previous studies conducted for *Pseudonomas* sp. and chlorpyrifos degradation, 1316 proteins were identified, representing 21% of the predicted proteome [47]. For *Priestia* sp., associated with the metabolism of the herbicide mesotrione, 1820 proteins were identified, representing 32.5% of the predicted proteome [48]. Therefore, the results obtained in this study are remarkably similar to those in previous reports. To the best of our knowledge, the proteome associated with pesticide degradation for *Achromobacter* and *Stenotrophomonas* has not been reported. Previous studies have shown that some bacterial proteins that degrade contaminants could be constitutive or inducible. For example, Zhao et al. [39] reported that the activity of *Bacillus cereus* GW-01 was inducible on β-cypermethrin metabolism. Similarly, Aswathi et al. [47] reported the induction of specific proteins in *Pseudomonas nitroreducens* AR-3 as being involved in chlorpyrifos degradation. Our results show that IPR exposure induced 25 to 111 proteins for 5 of the 6 strains, except *Achromobacter* sp. C1 with the most highly expressed proteins (445 solely expressed proteins) compared to the untreated control, which implies a response to IPR exposure. In a study conducted for *Burkholderia zhejiangensis* CEIB S4-3, during the methyl parathion degradation process, the overexpression of proteins related to the defense against oxidative stress and degradation of the pesticide was observed [49]. 

Both metabolic processes and cellular processes were mainly altered in *Achromobacter* sp. C1 after IPR exposure, probably due to the diverse substrate oxidation and dissimilation reaction by which substrate molecules are broken down as a function in bacteria to generate energy [50]. A bacterial cell is a highly specialized energy transformer, and different enzymatic reactions are responsible for supporting and maintaining life processes within the cell, such as transferase, oxidoreductase, and hydrolase enzymes, as reported in this study. Catalytic mechanisms to transform a pesticide through oxidation, reduction, or hydrolysis to generally produce a more water-soluble and less toxic product than the original compound are related to the previously mentioned enzymes [51]. *Achromobacter* spp. have been reported as IPR degraders [13,15] and IPR removers, and other dicarboximide pesticides have been associated with amidohydrolase enzymes (amidase) that are able to degrade these pesticides. In our study, analysis of the significantly upregulated proteins using the KEGG database identified 4 of a total of 17 metabolic pathways that might aid *Achromobacter* sp. C1 in tolerating and degrading IPR. A study by Aswathi et al. [47] reported that the major proteins differentially expressed in *P. nitroreducens* AR-3 exposed to chlorpyrifos were those involved in the biosynthesis of secondary metabolites and metabolism in various environments, which is consistent with our results (Appendix A).

Amidases or amidohydrolase enzymes constitute a large group of enzymes that can hydrolyze the C-N bond of various amide compounds, including pesticides [17]. Pasquarelli et al. [52] reported that a sequential reaction catalyzed by nitrile hydratase and amidase enzymes present naturally in *Microbacterium imperiale* CBS 498-74 was able to induce bromoxynil biotransformation into the corresponding acid. Yang et al. [33] reported a novel amidase named AceAB, purified and characterized from the acetamiprid-degrading strain *Pigmentiphaga* sp. strain D-2, that was able to hydrolyze the C-N bond of acetamiprid. Campos et al. [12] discussed the possible association of amidohydrolases and the cleavage of IPR. Yang et al. [23] reported a novel ipaH gene encoding an amidase responsible for the initial degradation step of IPR in *Paenarthrobacter* sp. strain YJN-5, while Zhang et al. [41] reported a hydrolase gene, duaH, which is responsible for cleavage of the urea side chain of 3,5-dichlorophenylurea acetic acid, thus yielding 3,5-DCA as the end product. 

In this study, different proteins with amidohydrolase function were elevated, lowered, or solely expressed in response to mechanisms implicated in IPR degradation and resistance by the strain *Achromobacter* sp. C1. Comparatively, downregulated FOR enzymes largely exceeded their presence in the metabolic changes observed for strain C1. In general, FOR can be described as an aliphatic amidase with restricted substrate specificity, as it only hydrolyzes formamide in the presence of water. FOR (BRENDA: EC3.5.1.49) is present in viruses and cellular organisms and acts in the pathway of aerobic L-tryptophan degradation to anthranilate (anthranilate pathway) in several bacteria; an example of this is the kynurenine formamidase [53] described by Kurnasov et al. [54]. According to our results, the main negative effects of IPR on strain C1 involve nicotinamide metabolism and benzoate degradation in the tryptophan metabolism, as a consequence of direct inhibition in the nicotinamide pathway but also indirectly due to the effect on FOR in phenylacetate, which is related as an intermediary in both phenylalanine metabolism and the pathway of styrene degradation. Nicotinamide is a compound with a diverse biological function, serving as a precursor of the coenzymes NAD+ and NADP+ related to glycolysis, respiration, and fatty acid synthesis [55]. Primary metabolic pathways, such as the metabolism of carbohydrates and fatty acids for energy production, are needed to stabilize a new homeostasis in the resistant bacteria [56]. Therefore, the inhibition of FOR proteins could be counteracted, considering that other proteins represented by CYC are upregulated in the presence of IPR. Specifically, a CYC with arylformamidase (EC3.5.1.9) activity participates in tryptophan metabolism, hydrolyzing formylkynurenine to generate kynurenine, which is a precursor of anthranilate and nicotinamide; this could be a metabolic mechanism of strain C1 to produce energy for the maintenance of bacterial growth under stress conditions [56]. Other upregulated proteins include AMDs, in the family of hydrolases, which are known to be important in the phase I metabolism of xenobiotics, including pesticides [57]. Zhao et al. [58] reported two new AMDs involved in the biodegradation of the pyridinecarboxamide insecticide flonicamid by *Microvirga flocculans*. AMD signature enzymes are a large group of hydrolytic enzymes found in both prokaryotes and eukaryotes. The role of these enzymes is the hydrolysis of amide bonds (CO-NH_2_), although the family has diverged widely with regard to substrate specificity and function [59]. Subunit A of Glu-tRNA (Gln) amidotransferase, which, in our study, was expressed solely on IPR treatment, is a heterotrimeric enzyme that catalyzes the formation of Gln-tRNA(Gln) by the transamidation of misacylated Glu-tRNA(Gln) via the amidolysis of glutamine [59]. Similarly, in a study conducted by Pankaj et al. [56], a Glu-tRNA (Gln) amidotransferase was overexpressed in cypermethrin-induced *Bacillus thuringiensis* strain SG4. The function of aminoacyl-tRNA synthesis is the attachment of an amino acid to the corresponding tRNA by an aminoacyl-tRNA synthetase. In this context, the amino acids glutamate and aspartate could act as substrates to synthesize proteins and other essential substances required by the strain *Achromobacter* sp. C1 to relieve the toxicity of degradation products of IPR, as observed during the degradation process of chlorpyrifos by *Bacillus megaterium* strain RRB [55]. Nevertheless, other enzymes like N-ethylammeline chlorohydrolase, N-formylglutamate deformylase, and N-formylglutamate amidohydrolase have not been previously identified in the metabolic pathway of strain C1, and the overexpression in response solely to the presence of IPR could indicate some responsibility in the IPR degradation process. N-ethylammeline chlorohydrolase acts in the pathway of atrazine biodegradation, transforming the metabolite deisopropylatrazine to deisopropylhydroxyatrazine [60]. According to the existing literature [13,32,40,41], the IPR degradation pathway to 3,5-DCA does not involve the dechlorination processes. However, in the breakdown of 3,5-DCA to chlorocatechol that ends in aniline, a hydrolase of this type could be involved, but further studies are required to confirm this. Finally, N-formylglutamate amidohydrolase, also known as N-formylglutamate deformylase (BRENDA:EC3.5.1.68), can act on carbon–nitrogen bonds other than peptide bonds in linear amides [53]. 

## 5. Conclusions

New bacteria able to degrade the fungicide IPR were isolated from a pesticide biopurification system, confirming that this system is a good source of microorganisms able to tolerate pesticides. Based on nano-LC-MS/MS, between 1057 and 1740 total identified proteins were present in the six isolates. Among them, 25 to 111 proteins were solely expressed in the presence of IPR, highlighting *Achromobacter* sp. C1 with 445 proteins caused by the induction of specific proteins/pathways that aid IPR degradation. The IPR on strain C1 involved metabolic pathways and cellular processes; in addition, molecular functions such as transferase, oxidoreductase, and hydrolase activity were identified. The most differentially expressed proteins in strain C1 were those involved in the metabolic pathways of biosynthesis of secondary metabolites and microbial metabolism in diverse environments. This work involved several amidases in six metabolic pathways, including phenylalanine metabolism, tryptophan metabolism, styrene degradation, aminobenzoate degradation, caprolactam degradation, and aminoacyl-tRNA biosynthesis. While formamidase was inhibited, other amidases were overexpressed, minimizing the effect of IPR on the metabolism of strain C1 and then favoring IPR degradation. Subunit A of Glu-tRNA (Gln) amidotransferase related to aminoacyl-tRNA synthesis expressed solely due to IPR’s presence could indicate that amino acids such as glutamate and aspartate act as substrates for the synthesis of proteins required by strain C1 to tolerate and degrade IPR. Other proteins such as N-ethylammeline chlorohydrolase were also identified. Future assays must associate certain proteins with IPR degradation beyond its main metabolite 3,5-DCA. However, these proteins could be confirmed by other strategies such as Western blot and targeted proteomics. In addition, protein fractions could be placed in the presence of IPR to observe whether there is degradation, with this degradation related to the enzymes present in the fraction. Few studies have detailed the proteomic changes that assist in the degradation of pesticides. Previously, we proposed *Achromobacter* sp. C1 isolated from a pesticide biopurification system as a promising bacterium able to degrade pesticides. In this study, we concluded that, due to its proteomic profile, the strain constitutes a biotechnological tool for treating IPR and likely other dicarboximide fungicides. 

## Figures and Tables

**Figure 1 microorganisms-11-02367-f001:**
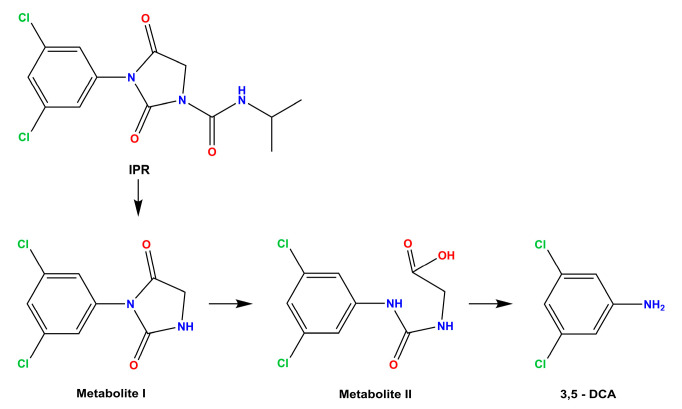
Overview of iprodione (IPR) degradation pathway. Metabolite I: 3,5-dichlorophenyl-carboxamide, and metabolite II: 3,5-dichlorophenylurea-acetate.

**Figure 2 microorganisms-11-02367-f002:**
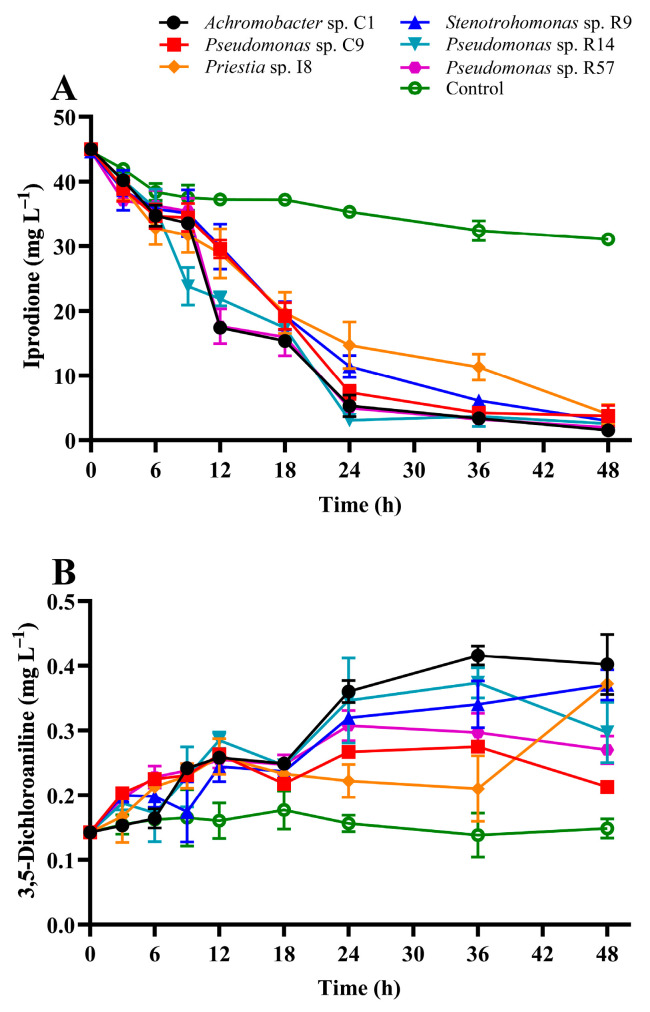
Iprodione removal (**A**) and 3,5-DCA production (**B**) in diluted LB medium supplemented with 50 mg L^−1^ of iprodione by biopurification system bacterial strains. The error bars represent the standard error of the means of three replicates.

**Figure 3 microorganisms-11-02367-f003:**
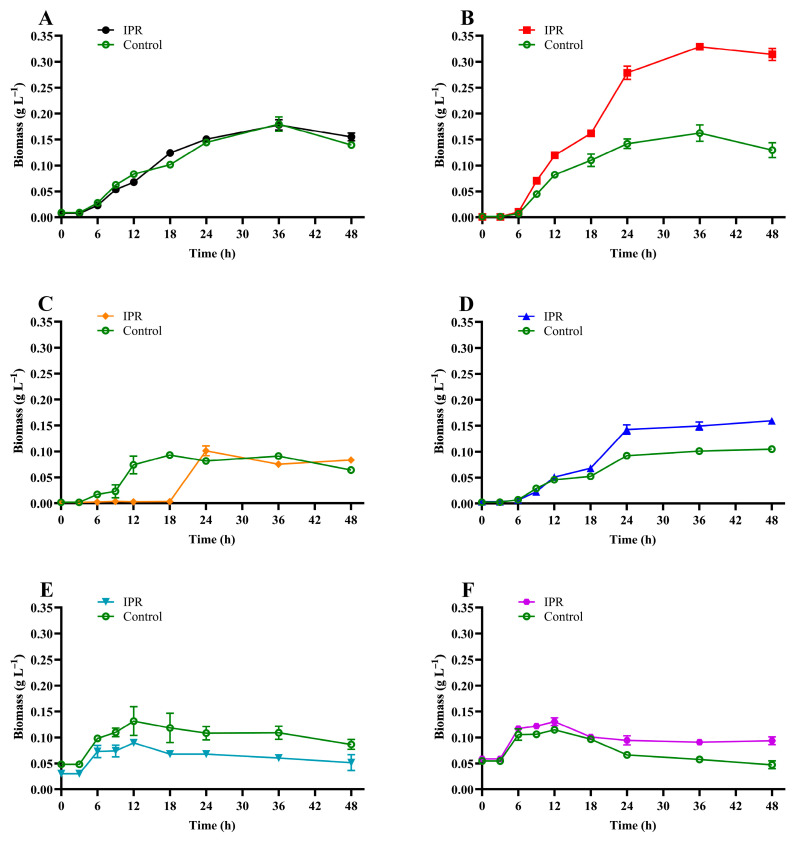
Growth of (**A**) *Achromobacter* sp. C1, (**B**) *Pseudomonas* sp. C9, (**C**) *Priestia* sp. I8, (**D**) *Stenotrophomonas* sp. R9, (**E**) *Pseudomonas* sp. R14, and (**F**) *Pseudomonas* sp. R57 in response to iprodione treatment. The error bars represent the standard error of the means of three replicates.

**Figure 4 microorganisms-11-02367-f004:**
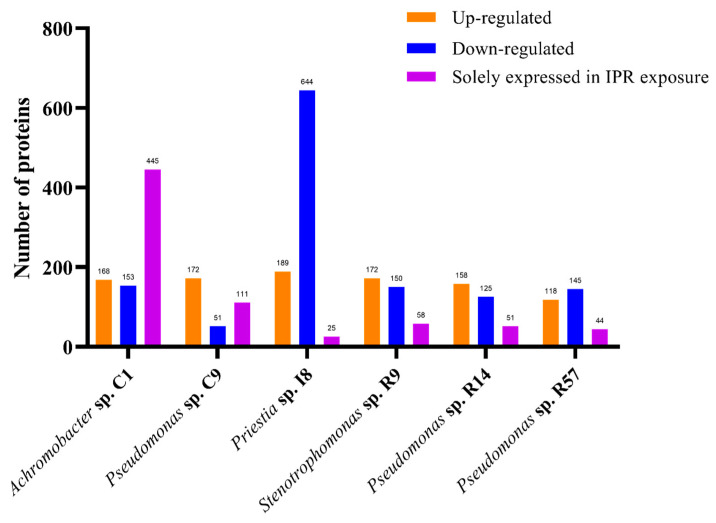
Differential expression of proteins by biopurification system bacteria *Achromobacter* sp. C1, *Pseudomonas* sp. C9, *Priestia* sp. I8, *Stenotrophomonas* sp. R9, *Pseudomonas* sp. R14, and *Pseudomonas* sp. R57 in response to iprodione (IPR).

**Figure 5 microorganisms-11-02367-f005:**
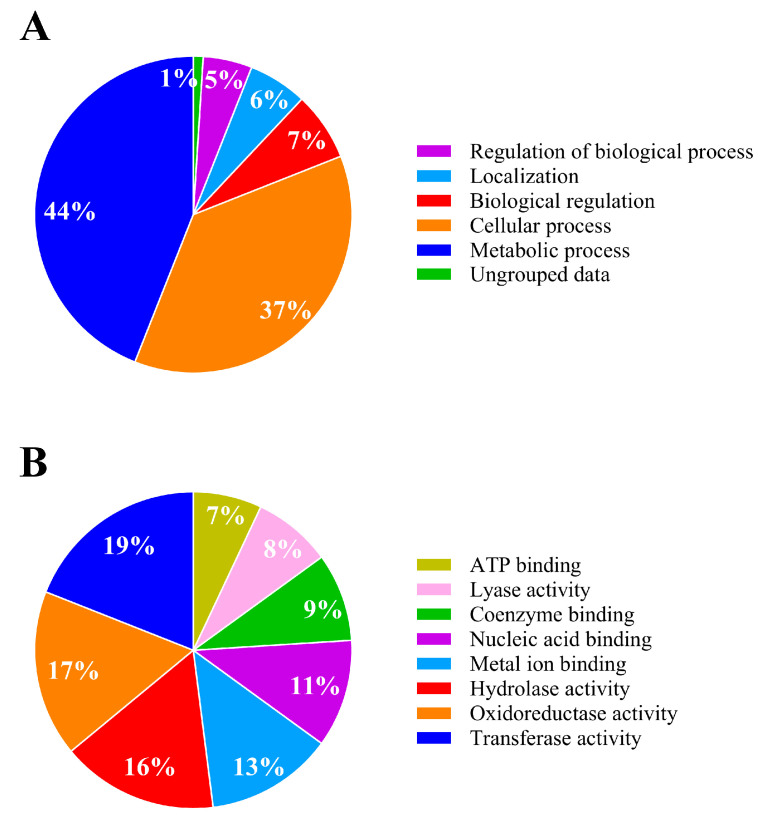
Metabolic activity analysis of *Achromobacter* sp. C1 in response to iprodione. (**A**) Bacterial biological process; (**B**) bacterial molecular function. Modified from Blast2GO software [30].

**Figure 6 microorganisms-11-02367-f006:**
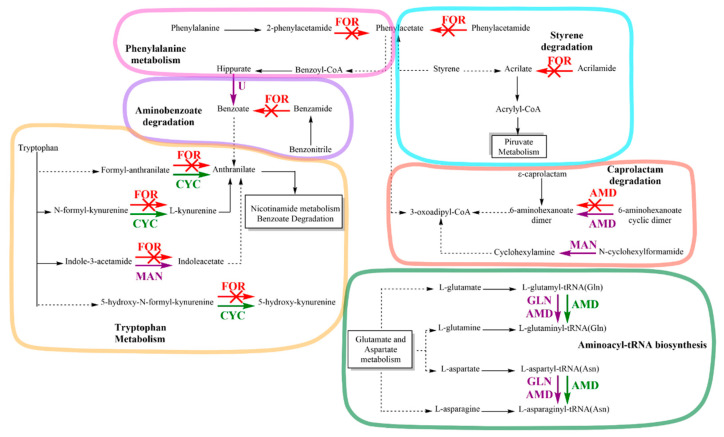
Scheme representing the metabolic changes observed for strain *Achromobacter* sp. C1 based on protein differential expression: (**→**) upregulated protein, (**→**) downregulated protein, and (**→**) solely expressed on IPR exposure. Up- and downregulated proteins are displayed in green and red with and X, respectively; those solely expressed on IPR exposure are displayed in purple. FOR, formamidase; U, uncharacterized hydrolase; CYC, cyclase; MAN, mandelamide hydrolase; AMD, amidase. GLN, glutamyl-tRNA(Gln) amidotransferase subunit A.

**Table 1 microorganisms-11-02367-t001:** First-order kinetics data for iprodione (IPR) removal, specific growth rate (µ), and biomass duplication time of biopurification system bacteria *Achromobacter* sp. C1, *Pseudomonas* sp. C9, *Priestia* sp. I8, *Stenotrophomonas* sp. R9, *Pseudomonas* sp. R14, and *Pseudomonas* sp. R57 in diluted LB liquid medium supplemented with an initial IPR concentration of 50 mg L^−1^ and evaluated over 48 h.

Parameters	C1	C9	I8	R9	R14	R57
Regression equation	−6.001x + 51.809	−5.623x + 52.203	−4.895x + 49.670	−5.482x + 52.288	−5.831x + 50.658	−5.908x+ 51.450
*K* (h^−1^)	0.076	0.073	0.047	0.060	0.067	0.073
T_1/2_ (h^−1^)	9.1	9.5	14.6	11.5	10.3	9.5
R^2^	0.960	0.948	0.982	0.962	0.955	0.941
Specific growth rate (µ)	0.140	0.092	0.299	0.123	0.034	0.018
Biomass duplication time (h)	4.94	7.55	2.32	5.63	20.24	39.12

**Table 2 microorganisms-11-02367-t002:** *Achromobacter* sp. C1 identified amidase proteins differentially expressed under 50 mg L^−1^ iprodione exposure.

Accession Number	Protein Name	Fold	Function
Upregulated proteins			
A0A3Q9KM03 A0A3Q9KKP8	Aspartyl/glutamyl-tRNA(Asn/Gln) amidotransferase subunit C Amidase	3.6 1.9	asparaginyl-tRNA synthase (glutamine-hydrolyzing) activity, glutaminyl-tRNA synthase (glutamine-hydrolyzing) activity, transferase activity, ligase activity, ATP binding, nucleotide binding Amidase and indoleacetamide hydrolase activity
A0A2S0I6B2	Cyclase	1.6	Arylformamidase activity
Downregulated proteins			
A0A2S0I9W4	Formamidase	2.6	Hydrolase activity, acting on carbon–nitrogen (but not peptide) bonds, in linear amides
A0A2S5GVY3	Amidase	2.7	Hydrolase activity
Expressed solely on IPR treatment proteins			
A0A3S4NGC5	Glutamyl-tRNA(Gln) Amidotransferase subunit A	Unique	ATP binding, glutaminyl-tRNA synthase (glutaminehydrolyzing) activity, hydrolase activity and transferase activity
A0A3S4PMM6	Glutamyl-tRNA(Gln) amidotransferase subunit A	Unique	Hydrolase activity, ligase activity and transferase activity
A0A2S5GXA3	Glutamyl-tRNA(Gln) amidotransferase subunit A	Unique	ATP binding, glutaminyl-tRNA synthase (glutaminehydrolyzing) activity, hydrolase activity and transferase activity
A0A3Q9KJB0	Amidase	Unique	Hydrolase activity
A0A2S5GNM1	Amidase	Unique	Hydrolase activity
A0A448CBV6	Uncharacterized hydrolase YxeP	Unique	Hydrolase, amidohydrolase activity
A0A3S4Q6N4	Uncharacterized hydrolase YxeP	Unique	Hydrolase, amidohydrolase activity
A0A2S5GW20	N-ethylammeline chlorohydrolase	Unique	Hydrolase activity, amidohydrolase-related
A0A2S5GLS7	N-formylglutamate deformylase	Unique	N-formylglutamate deformylase, n-formylglutamate amidohydrolase activity
A0A2S0I4Y1	N-formylglutamate amidohydrolase	Unique	Hydrolase, N-formylglutamate amidohydrolase activity
A0A448C6N6	Mandelamide hydrolase	Unique	Mandelamide amidase activity
A0A3S9YT14	Amidase	Unique	Hydrolase activity
A0A3S9YSH8	Amidohydrolase	Unique	Hydrolase, amidohydrolase activity
A0A2S5GNK0	Amidohydrolase	Unique	Hydrolase, amidohydrolase activity

## Data Availability

Not applicable.

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
