# Peer review of "Metabolic Profiling and Comparative Proteomic Insight in Respect of Amidases during Iprodione Biodegradation"

_microorganisms, 2023, doi:10.3390/microorganisms11102367_

Round 1
Reviewer 1 Report
The authors isolated and identified several strains capable of degrading the fungicide iprodione. They tested these isolates, as well as isolates obtained from a previous study in terms of biodegradation kinetics, growth and most importantly, the changes in their protein expression profiles during iprodione degradation, particularly focusing on strain Achromobacter sp. C1 and specifically paying special attention to amidases, which are probably involved in the degradation pathway and in other metabolic processes associated with the exposure to iprodione.
This dynamic proteomic analysis is the strongest point in the manuscript and is seldom carried out in other xenobiotic degradation studies due to the technical and analytical challenges associated with it, and therefore is a very valuable asset. Insights gained through this approach are extensively presented in the results and discussion sections and provide valuable information to researchers interested in iprodione degradation and proteomic analysis for xenobiotic removal in general.
Specific comments
L36. The second sentence in the introduction refers to "fungal-resistant"; should be "fungicide-resistant".
L36. Genera should be italicized. This should be checked and corrected in several parts of the manuscript.
L55. Start a new paragraph when introducing the concept of Biopurification Systems, to improve readability.
L173. How exactly was the biomass measured?
L180. The limit of quantification for iprodione is missing.
L301: Should be "Pseudomonas vancouverensis"
L335 to 341. It is interesting that Achromobacter C1 was the most efficient degrader for the parent compound, but at the same time it also produces the highest amount of the toxic intermediate 3,5 DCA at the end of the assay. What implications can this have in terms of detoxifying the matrix? A brief mention of this observation in the discussion is recommended.
L640. What sort of approaches would the authors suggest to inequivocally associate the proteins to IPR degradation?
Author Response
Comments Reviewer 1 |
Response to reviewer 1 |
L36. The second sentence in the introduction refers to "fungal-resistant"; should be "fungicide-resistant". |
Sentence was changed (page 1) |
L36. Genera should be italicized. This should be checked and corrected in several parts of the manuscript. |
It was corrected. |
L55. Start a new paragraph when introducing the concept of Biopurification Systems, to improve readability. |
It was corrected (Line 66) |
L173. How exactly was the biomass measured? |
This was clarified by incorporating the following sentence into the text: “Biomass growth was quantified by absorbance measurement at 600 nm and converted to cell dry weight (g L-1) through a calibration curve [27].” (Line 190-192) |
L180. The limit of quantification for iprodione is missing |
LOQ was added in the text (Line 195) |
L301: Should be "Pseudomonas vancouverensis" |
It was corrected (Line 318) |
L335 to 341. It is interesting that Achromobacter C1 was the most efficient degrader for the parent compound, but at the same time it also produces the highest amount of the toxic intermediate 3,5 DCA at the end of the assay. What implications can this have in terms of detoxifying the matrix? A brief mention of this observation in the discussion is recommended. |
A paragraph to explain this was incorporated in the discussion (Lines 533-542) |
L640. What sort of approaches would the authors suggest to inequivocally associate the proteins to IPR degradation? |
We think the answer would be another paper, independent of this screening article. Proteomics has suggested a number of deregulated proteins, including amidases. The experimental design, strains that degrade and do not degrade IPR, in the presence and absence of IPR, revealed the proteins that are possibly involved in IPR degradation. “These proteins could be confirmed by other strategies such as western blot and targeted proteomics. Also, protein fractions could be placed in the presence of IPR and see if there is degradation, relating it to the enzymes present in the fraction”. These statements were incorporated into the text. (Line 686-689). |
Reviewer 2 Report
Iprodione (IPR) was banned due to its toxicity, and it is a significant work to biodegrade IPR. I recommend it publication in Microorganisms after major revision.
1) The IPR biodegradation pathway should be represented in main text, which will let readers understand well.
2) Mass balance between IPR and 3,5-DCA should be done, e.g. authours should calculate how many 3,5-DCA generated with IPR disappearance, for which readers would know well how many IPR was transformated into orther intermediates
3) Corresponding to Figures 1 and 2, authors did not describe clearly in M&M, for which I suggested authors perfect them.
4) For Comparative Proteomic Study, I suggest authors should combine the results with data in Figures 1 and 2. Current analysis did not give readers clear understand to the article.
5) Figure 3 should be discussed more, because current discussion was superficial.
The manuscript sould be moderated in English.
Author Response
Comments Reviewer 2 |
Response to reviewers 2 |
The IPR biodegradation pathway should be represented in main text, which will let readers understand well. |
A general IPR biodegradation pathway was incorporated in the introduction. (Line 52-65) |
Mass balance between IPR and 3,5-DCA should be done, e.g. authours should calculate how many 3,5-DCA generated with IPR disappearance, for which readers would know well how many IPR was transformated into orther intermediates |
The following statement was added: “In relation to the initial mass of IPR, the amount in mass of 3,5-DCA represented on aver-age 1.40% of the parent compound at the end of the study. Thus, the remaining percentage could be represented by the IPR intermediates identified as metabolite I, metabolite II, or even glycine, which is produced together with 3,5-DCA in the final stage of IPR degrada-tion. However, more analyses are required to clarify this.” (Line 528-533) |
Corresponding to Figures 1 and 2, authors did not describe clearly in M&M, for which I suggested authors perfect them. |
It was corrected by improving the description of biomass determination. |
For Comparative Proteomic Study, I suggest authors should combine the results with data in Figures 1 and 2. Current analysis did not give readers clear understand to the article. |
We tried to combine figures 1 and 2, but the results were confusing. However, to clarify that the proteomics study was performed for a specific time of the degradation study we linked Fig. 2 and Fig. 3 in the text (Line 281-382). |
Figure 3 should be discussed more, because current discussion was superficial. |
Discussion was improved and new reference was added [line 560-579]. |
Please, see the attached document.

Reviewer 3 Report
In this manuscript, authors isolated four new bacteria tolerant to IPR from a biopurification system designed to treat pesticides. A study of biodegradation using these four strains with two previously isolated strains was comparatively evaluated. Comparative proteomic study was performed with these strains. Analysis of differentially expressed proteins and amidase protein expression on amidase metabolic pathways in response to IPR were performed for strain Achromobacter sp. C1.
I don’t think this work has enough interest to be published in “Microorganisms”. First, the manuscript showed a lot of work about isolation of four new bacteria and evaluation of their biodegradation abilities of IPR. However, the best degradation rate of IPR was presented by Achromobacter sp. C1, which was a strain isolated previously by authors. And the proteomic study also focused on this previously isolated strain. So, what’s the meaning of showing isolation of IPR tolerant bacteria from BPS in this manuscript? The authors already reported the isolation of bacteria from BPS, such as Achromobacter sp. C1, which could degrade the IPR. Next, the authors focused on the analysis of differentially expressed amidase of Achromobacter sp. C1, but the results were confusing since some amidases were up regulated and some amidases were down regulated. Then, the authors tried to explain the proteomic data and did the analysis of amidase protein expression on amidase metabolic pathways in response to IPR, but I don’t think I can get clear conclusions or speculates from these analyses about the biodegradation pathways of IPR by this strain. The authors should make more in-depth analysis of the proteomic data and related pathways and put forward the clear views and hypotheses about the biodegradation pathways of IPR by Achromobacter sp. C1.
Moderate editing of English language required
Author Response
I don’t think this work has enough interest to be published in “Microorganisms”. First, the manuscript showed a lot of work about isolation of four new bacteria and evaluation of their biodegradation abilities of IPR. However, the best degradation rate of IPR was presented by Achromobacter sp. C1, which was a strain isolated previously by authors. And the proteomic study also focused on this previously isolated strain. So, what’s the meaning of showing isolation of IPR tolerant bacteria from BPS in this manuscript? The authors already reported the isolation of bacteria from BPS, such as Achromobacter sp. C1, which could degrade the IPR. Next, the authors focused on the analysis of differentially expressed amidase of Achromobacter sp. C1, but the results were confusing since some amidases were up regulated and some amidases were down regulated. Then, the authors tried to explain the proteomic data and did the analysis of amidase protein expression on amidase metabolic pathways in response to IPR, but I don’t think I can get clear conclusions or speculates from these analyses about the biodegradation pathways of IPR by this strain. The authors should make more in-depth analysis of the proteomic data and related pathways and put forward the clear views and hypotheses about the biodegradation pathways of IPR by Achromobacter sp. C1. |
The search for specialized microorganisms to degrade complex compounds is required for the development of biotechnological tools to be used in degradation systems. We searched for new strains in a purification system that are tolerant and able to degrade the contaminant under study obtaining good results. However, when comparing with previously isolated strains the results were better, so we decided to perform the proteomic study despite being a strain previously isolated and characterized to degrade different pesticides. The manuscript clearly mentions that two previously isolated strains were incorporated for comparative purposes.
In addition, we think the answer would be another paper, independent of this screening article. Proteomics has suggested a number of deregulated proteins, including amidases. The experimental design, strains that degrade and do not degrade IPR, in the presence and absence of IPR, revealed the proteins that are possibly involved in IPR degradation. These proteins could be confirmed by other strategies such as western blot and targeted proteomics. Also, protein fractions could be placed in the presence of IPR and see if there is degradation, relating it to the enzymes present in the fraction. These statements were incorporated into the text as projection for future assay.
|

Round 2
Reviewer 2 Report
Authors revised their manuscript according to comments from reviewers, and it reaches the level for publication.
Reviewer 3 Report
Even though I still can't completely agree with the authors’ response, the proteomic data analysis part was improved in the revised version. Through further reviewing the manuscript, I think the proteomic data analysis could be useful to reveal the metabolic pathways and proteins that are possibly involved in IPR degradation of Achromobacter sp. C1 and other IPR-degrading microorganisms. That has potential interest in the related research fields. So, I would suggest accepting the current manuscript.